# NeuroGF: A Neural Representation for Fast Geodesic Distance and Path Queries

**Qijian Zhang[1], Junhui Hou[1]*, Yohanes Yudhi Adikusuma[2], Wenping Wang[3], Ying He[2]**

[1]Department of Computer Science, City University of Hong Kong, Hong Kong SAR, China
[2]School of Computer Science and Engineering, Nanyang Technological University, Singapore
[3]Department of Computer Science and Engineering, Texas A&M University, Texas, USA
`qijizhang3-c@my.cityu.edu.hk, jh.hou@cityu.edu.hk`
`Yohanes.adikusuma@ntu.edu.sg, wenping@tamu.edu, yhe@ntu.edu.sg`

## Abstract

Geodesics play a critical role in many geometry processing applications. Traditional algorithms for computing geodesics on 3D mesh models are often inefficient and slow, which make them impractical for scenarios requiring extensive querying of arbitrary point-to-point geodesics. Recently, deep implicit functions have gained popularity for 3D geometry representation, yet there is still no research on neural implicit representation of geodesics. To bridge this gap, we make the first attempt to represent geodesics using implicit learning frameworks. Specifically, we propose neural geodesic field (NeuroGF), which can be learned to encode all-pairs geodesics of a given 3D mesh model, enabling to efficiently and accurately answer queries of arbitrary point-to-point geodesic distances and paths. Evaluations on common 3D object models and real-captured scene-level meshes demonstrate our exceptional performances in terms of representation accuracy and querying efficiency. Besides, NeuroGF also provides a convenient way of jointly encoding both 3D geometry and geodesics in a unified representation. Moreover, the working mode of per-model overfitting is further extended to generalizable learning frameworks that can work on various input formats such as unstructured point clouds, which also show satisfactory performances for unseen shapes and categories. Our code and data are available at `https://github.com/keeganhk/NeuroGF`.

## 1  Introduction

The computation of geodesic distances and paths on 3D mesh models has been extensively studied over the past few decades and plays a critical role in many geometry processing tasks, including texture mapping, shape description, correspondence, and deformation. Traditional computational geometry methods, such as discrete wavefront propagation [25, 8, 39, 41, 32] and geodesic graphs [46, 2], excel at computing exact or high-quality approximate geodesic distances and paths on meshes with arbitrary triangulation. However, these methods often suffer from computational inefficiency or require a pre-computation step. On the other hand, partial differential equation (PDE) methods [17, 10, 36] are renowned for their efficiency, flexibility and ease of implementation, but the accuracy of their results is sensitive to the quality of mesh triangulation.

It is also worth mentioning that existing geodesic algorithms are primarily designed for computing single-source all-destination (SSAD) or multiple-sources all-destination (MSAD) geodesics on meshes. While there are some works focused on all-pairs geodesic computations, such as [40, 4, 12], these methods are built upon relatively complex computational and optimization processes.

---

*Corresponding author

37th Conference on Neural Information Processing Systems (NeurIPS 2023).

Dealing with the trade-off among accuracy, speed, and complexity for geodesic computation remains technically non-trivial.

As 3D deep learning continues to advance, integrating geodesic information into deep learning frameworks becomes increasingly important for a wide range of applications, enabling more accurate and efficient analysis and manipulation of complex 3D shapes. However, until now, none of the existing geodesic algorithms have been seamlessly integrated into deep neural networks.

In this paper, we make the first attempt to utilize deep neural networks for accurately and efficiently answering geodesic distance and path queries. Towards this goal, we propose neural geodesic fields (NeuroGFs), an overfitting paradigm for representing geodesic distances and paths of a given 3D shape in a single neural model in an implicit fashion. Our approach is highly desirable to applications that require extensive and frequent online queries of point-to-point geodesic computations. Evaluations on common 3D models demonstrate that NeuroGFs excel in solving both SSAD and point-to-point geodesic problems. They achieve exceptional performance and strike a satisfactory balance between representation accuracy, model compactness, and query efficiency. Specifically, NeuroGFs are capable of achieving computation times of less than 1 ms for solving SSAD on 3D meshes consisting of 400K vertices, surpassing traditional methods, such as the heat method and discrete geodesic graphs, by two orders of magnitude. Besides, NeuroGFs offer the unique advantage of encoding both 3D geometry and geodesics in a unified representation. It is important to note that despite its compact size, with only 259K hyperparamters, the NeuroGF network consistently delivers high accuracy, with a relative mean error of less than 0.5%. Moreover, to facilitate more diverse applications, we further investigate generalizable learning frameworks of NeuroGFs by introducing different types of 3D shape feature encoders, which can well generalize to unseen shapes and novel categories. These results strongly demonstrate that NeuroGF is an effective and efficient tool for geodesic queries on 3D meshes, opening up exciting possibilities for a wide range of potential applications in the field of 3D deep learning. The main contributions of this work can be summarized as follows:

- We make the first attempt to explore neural implicit representations for answering geodesic queries on 3D meshes.

- In addition to geodesic distances, we model shortest paths as discrete sequences of ordered 3D points and customize a learning process of feature-guided curve deformation, which can flexibly approximate the shortest path between any pair of source and target query points.

- We construct a unified representation structure for both geometric and geodesic information of given 3D shapes. Compared with traditional computational and optimization-based approaches, the proposed NeuroGF framework is easy to implement, highly efficient for online query, and naturally enjoys the powerful parallelism of modern GPUs.

- We evaluated the performances of our NeuroGF representation paradigms in both overfitted and generalizable working modes, demonstrating the great potential and technical extensibility of our approach.

## 2   Related Work

**Discrete Geodesics.** The existing techniques for computing geodesics on 3D surfaces can be broadly classified into six categories: computational geometry methods, graph methods, PDE methods, surface partitioning methods, and iterative methods.

Traditional computational geometry methods [25, 8, 39, 47, 41, 32] are capable of computing exact geodesic distances and paths on arbitrary meshes. However, these methods are computationally expensive, especially when dealing with large meshes. Additionally, while these methods are well-suited to solving the single-source/multiple-sources and all-destinations problem, they are not practical for computing all-pairs geodesic distances due to their high computational costs.

Graph-based methods [46, 2] exploit the local nature of the discrete geodesic problem by constructing a sparse graph, where each edge represents a short geodesic path. These paths are computed locally by using exact geodesic algorithms. By transforming the mesh-based geodesic problem into a graph-based shortest-path problem, graph-based methods can use efficient algorithms such as Dijkstra's algorithm to compute geodesic distances and paths. Furthermore, their results are guaranteed to be a true distance metric and have controllable accuracy. However, achieving high-accuracy geodesic

computation often requires significant memory for storing the graphs, as the complexity of the graph increases with the desired accuracy.

PDE methods are highly flexible and can work for a wide range of discrete domains, including point clouds, implicit functions, polygonal meshes, regular grids and even broken meshes. They solve the Eikonal equation either directly [17, 44, 5, 37] or indirectly [10, 3, 36]. PDE methods are efficient and easy to implement. However, since they compute only a first-order approximation of geodesic distances, they typically require meshes with fairly good tessellation. To obtain geodesic paths using PDE methods, the negative gradient of the computed geodesic distances must be back-traced. However, the accuracy of the resulting paths also depends on the mesh quality, as the gradient may not be continuous or may have sharp discontinuities at the mesh edges. Nevertheless, PDE methods remain popular for their efficiency and versatility in computing geodesics on various discrete domains, including those beyond polygonal meshes.

Surface partition methods aim to break down the discrete geodesic problem into small sub-problems. The geodesic tracks (GT) method [23] places evenly-spaced, source-independent Steiner points on edges. Given a source vertex, it constructs a Steiner-point graph that partitions the surface into mutually exclusive tracks. Within each triangle, the tracks form sub-regions with approximately linear change in the distance field, enabling efficient approximation of geodesic distances. While the GT method supports high-quality geodesic path and isoline tracing, it remains inefficient for point-to-point distance and path queries due to the need for constructing the Steiner point graph using a Dijkstra-like sweep. The separator-based method (SEP) [12] performs a nested bisection of the input mesh using separator curves. It approximates the distances between each mesh vertex and a small relevant subset of these curves using polynomials. Consequently, the geodesic distance between any two mesh vertices can be approximated by solving a small number of simple univariate minimization problems. The SEP method efficiently handles point-to-point distance queries but does not provide geodesic paths. Furthermore, it is limited to query points located on mesh vertices, and it may not perform well on anisotropic meshes.

Additionally, there are iterative approaches like the edge flipping method [33] and the curve shortening method [38] that specially target the computation of geodesic paths and loops. These methods typically rely on an initial path and proceed in an iterative manner. Their time complexities depend on the area of the region the path sweeps. In contrast, our method does not require an initial path and can efficiently compute geodesics with a single forward pass through the network. It provides flexibility in solving both SSAD/MSAD and point-to-point geodesics. Furthermore, our method takes advantage of the high parallelism offered by modern graphics hardware, resulting in impressive runtime performance.

**3D Deep Learning with Geodesics.** Geodesics provide valuable structural information that can be seamlessly integrated into deep learning frameworks for diverse tasks in shape analysis. Several existing works have leveraged geodesic information within deep learning models. For example, GCNN [22] introduces manifold-adapted convolutional operators, constructing local geodesic coordinate systems for patch extraction. MeshCNN [14] combines specialized convolution and pooling layers operating on mesh edges by leveraging intrinsic geodesic connections for performing feature extraction on triangular meshes. Geodesic-Former [28] proposes geodesic-guided transformer model for few-shot instance segmentation of 3D point clouds, where geodesic distance information is leveraged to tackle the issue of imbalanced point density. GSA [18] designs geodesic self-attention to facilitate capturing long-range geometrical dependencies of point cloud objects. In the above studies, geodesics are exploited as additional input information to boost the subsequent learning models. Differently, GeoNet [15] explicitly learns geodesics-informed representations from unstructured point clouds in a supervised manner, which can be considered as a pre-training process of the target backbone networks. More recently, GraphWalks [31] proposes a differentiable geodesic path estimator learned in a supervised manner, but its prediction accuracy is still unsatisfactory. In terms of working modes, [15, 31] targeted at learning generalizable deep models for geodesics prediction, while in this work we tend to explore both overfitted and generalizable learning frameworks. Concurrently, [49] proposes a graph-based geodesic learning network that works on mesh models, while our implicit field-based modeling paradigm shows better flexibility for various input formats such as unstructured point cloud data.

**Neural Implicit Representation for 3D Shape Geometry**. The working mechanisms of existing neural implicit representations can be categorized as generalizable [29, 24, 9, 30, 6, 16] and overfitted

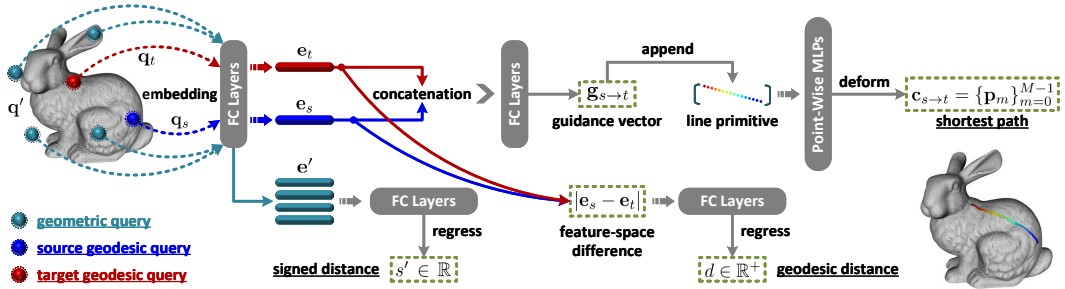

Figure 1: Overall workflow of querying point-to-point geodesic distances and shortest paths as well as extracting signed distance fields for surface geometry representation from NeuroGFs.

paradigms [34, 35, 27, 21, 26, 45, 19]. Essentially, the former category of approaches tends to learn generalizable shape priors from a large collection of shapes at the cost of a lower-fidelity representation, while the latter category of approaches aims at reproducing a single shape with high-fidelity by accordingly overfitting a single neural network model. In this paper, we start by following the representation paradigm of the latter overfitted neural implicit models, i.e., separately overfitting a single neural network on a single target mesh model, to validate the representation power of our NeuroGFs. Then, we further make efforts to extend generalizable schemes to learn geodesics for unseen shapes and novel categories. Generally, despite the proliferation of research on deep implicit geometric information representation, to the best of our knowledge, there is still no such effort on implicit geodesics representation.

## 3 Proposed Method

### 3.1 Problem Formulation

Denote by $\mathcal{S}$ a given geometric shape represented in the form of a 3D mesh model $\mathcal{M}_\mathcal{S} = (\mathcal{V}, \mathcal{E}, \mathcal{F})$, where $\mathcal{V}, \mathcal{E}, \mathcal{F}$ are the sets of vertices, edges, and faces, respectively. In an overfitted representation paradigm, our ultimate goal is to construct a neural model $\mathcal{N}_\Theta$ parameterized by network weights $\Theta$, which are particularly optimized on the given shape $\mathcal{S}$ for fitting geodesic information. Specifically, given an arbitrary pair of geodesic queries, we deduce their corresponding point-to-point geodesic distance and shortest path through $\mathcal{N}_\Theta$, which can be formulated as:

$$\{d, \mathbf{c}_{s \to t}\} = \mathcal{N}_\Theta(\mathbf{q}_s; \mathbf{q}_t), \tag{1}$$

where $\mathbf{q}_s$ and $\mathbf{q}_t$ denote the source and target 3D points located on the underlying surface of shape $\mathcal{S}$, $d \in \mathbb{R}^+$ is a scalar value representing the geodesic distance between the two geodesic query points, and $\mathbf{c}_{s \to t} \in \mathbb{R}^{M \times 3}$ represents the shortest path from source $\mathbf{q}_s$ to target $\mathbf{q}_t$ modeled as a discrete sequence of $M$ ordered 3D points, i.e., $\mathbf{c}_{s \to t} = \{\mathbf{p}_m\}_{m=0}^{M-1}$, where the subscript $m$ indicates the order index of each point.

In the offline training phase, we employ a supervised learning approach to overfit the neural network $\mathcal{N}_\Theta$ on a specific shape $\mathcal{S}$. The training is performed using a collection of ground-truth geodesics pre-computed from its corresponding 3D mesh $\mathcal{M}$. Once the overfitting process is completed, we can efficiently compute point-to-point geodesics for arbitrary online queries via a single forward pass through the trained network $\mathcal{N}_\Theta$.

### 3.2 Architectural Design

In principle, we stick to the most concise technical implementation of the overall learning pipeline, without highly introducing sophisticated modules or mechanisms, for exploring the viability and potential of our proposed NeuroGF representation paradigm. As illustrated in Figure 1, all required learnable components of $\mathcal{N}_\Theta$ are simply built upon the stack of either fully-connected (FC) layers or point-wisely shared multi-layer perceptrons (MLPs). The input query points are individually lifted to high-dimensional feature vectors using FC layers, enabling subsequent manipulation of the data. For ease of representation, the feature embeddings of a pair of source and target queries, $\mathbf{q}_s, \mathbf{q}_t \in \mathbb{R}^3$, are denoted as $\mathbf{e}_s, \mathbf{e}_t \in \mathbb{R}^{D_e}$. These embeddings are obtained by passing the inputs through three consecutive FC layers, progressively increasing the output channels. After obtaining the source and

target feature embeddings $\mathbf{e}_s$ and $\mathbf{e}_t$, we proceed by deploying two separate learning branches to predict the geodesic distance $d$ and the shortest path $\mathbf{c}_{s \rightarrow t}$.

In addition to the geodesic distance and shortest path learning branches, we also incorporate an auxiliary learning branch for surface geometry representation by predicting the corresponding signed distance field (SDF). Given a query point $\mathbf{q}' \in \mathbb{R}^3$ and its lifted $D_e$-dimensional feature embedding $\mathbf{e}'$, we predict its signed distance $s' \in \mathbb{R}$. This allows us to obtain a unified representation that encodes both geodesics and geometry. It is worth emphasizing that, during training, $\mathbf{q}'$ is sampled from the entire bounding sphere of the given shape $\mathcal{S}$, while $\mathbf{q}_s$ and $\mathbf{q}_t$ are selected from the mesh vertex set $\mathcal{V}$. During testing, $\mathbf{q}_s$ and $\mathbf{q}_t$ can be arbitrary points on the underlying shape surface, not limited to the vertices of the given mesh.

Overall, after the initial query point embedding, our neural network architecture consists of three parallel learning branches for predicting geodesic distances, shortest paths, and signed distance fields. While the three types of outputs are closely correlated, we deliberately design the intermediate learning processes in an inter-independent manner. This means that there is no explicit interaction (such as feature fusion or propagation) among the branches. We make this design choice to avoid redundant computations in scenarios where only one specific property is required for online query.

***Neural Fitting of Geodesic Distances.*** The geodesic distance regression is modeled as a learnable function of the "feature-space difference" between source and target queries. We compute an absolute difference vector between query embeddings $\mathbf{e}_s$ and $\mathbf{e}_t$, which is further mapped to a scalar value of the geodesic distance through a stack of FC layers, which can be formulated as:

$$d = \mathcal{B}_{\text{gdist}}(|\mathbf{e}_s - \mathbf{e}_t|), \tag{2}$$

where $| * |$ means element-wisely taking the absolute value of each vector entry, and the geodesic distance learning branch $\mathcal{B}_{\text{gdist}} : \mathbb{R}^{D_e} \rightarrow \mathbb{R}^+$ is implemented as three consecutive FC layers, where the output dimension of the last layer is set to 1.

***Neural Fitting of Shortest Paths.*** The shortest path generation is achieved as a feature-guided curve deformation process by adapting previous "folding-style" 3D shape reconstruction decoders [43, 13], where pre-defined 2D regular grid primitives conditioned on unique shape signatures are smoothly deformed to the target 3D shape surface. Inheriting the same working mechanism, we tend to deform the straight line segment between a pair of source and target geodesic query points to the 3D curve of their shortest path, which is conditioned on a learnable deformation guidance vector exported from source and target point embeddings.

Formally, we start by sequentially sampling $M$ discrete 3D points with uniform intervals along the directed straight line segment from source $\mathbf{q}_s$ to target $\mathbf{q}_t$, as given by:

$$\mathbf{l}_{s \rightarrow t} = \{\mathbf{q}_s + (\mathbf{q}_t - \mathbf{q}_s)/(M-1) \cdot m\}_{m=0}^{M-1}, \tag{3}$$

where $\mathbf{l}_{s \rightarrow t} \in \mathbb{R}^{M \times 3}$ serves as the initial line primitive to be deformed. In the online query phase, one can flexibly adjust the density of curve points according to the actual requirements by simply sampling the initial straight line segment with different uniform intervals. Next, the required curve deformation guidance vector can be deduced by:

$$\mathbf{g}_{s \rightarrow t} = \hbar([\mathbf{e}_s; \mathbf{e}_t]), \tag{4}$$

where $\mathbf{g}_{s \rightarrow t} \in \mathbb{R}^{D_e}$ uniquely determines the deformation result of the corresponding $\mathbf{l}_{s \rightarrow t}$, $[*; *]$ denotes channel concatenation, $\hbar : \mathbb{R}^{D_e + D_e} \rightarrow \mathbb{R}^{D_e}$ represents a learnable mapping function that can be simply implemented as a single FC layer.

To perform feature-guided curve deformation, we append $\mathbf{g}_{s \rightarrow t}$ to each point in the initial line primitive $\mathbf{l}_{s \rightarrow t}$. The resulting $M$-by-$(D_e + 3)$ feature matrix is row-wisely embedded for shortest path generation, which can be formulated as:

$$\mathbf{c}_{s \rightarrow t} = \mathcal{B}_{\text{spath}}([\mathbf{g}_{s \rightarrow t}; \mathbf{l}_{s \rightarrow t}]), \tag{5}$$

where $\mathcal{B}_{\text{spath}}$ comprises four point-wise MLPs with the output dimension of the last layer set to 3.

***(Auxiliary) Neural Fitting of Signed Distances.*** As discussed before, there is an auxiliary learning branch that consumes the feature embedding $\mathbf{e}'$ of the given geometric query point $\mathbf{q}'$ as input and regresses a scalar value of its signed distance, which can be formulated as:

$$s' = \mathcal{B}'_{\text{sdist}}(\mathbf{e}'), \tag{6}$$

where $\mathcal{B}'_{\text{sdist}} : \mathbb{R}^{D_e} \rightarrow \mathbb{R}$ is implemented as three consecutive FC layers with the output dimension of the last layer set to 1.

## 3.3 Learning Objective

Corresponding to the three types of outputs, the overall learning objective of NeuroGF representations comprises three loss function terms for the supervision of predicted geodesic distances $d$, curve points of shortest paths $\mathbf{c}_{s \to t}$, and signed distances $s'$. Additionally, we also impose two auxiliary constraint terms on the generated shortest paths to enhance the consistency of the three output shape properties. Specific mathematical formulations are given as follows.

*1) Supervision of Geodesic Distances.* We compute $L_1$ loss between the predicted and ground-truth geodesic distances $d$ and $\tilde{d}$ as:

$$\ell_{\text{gdist}} = \left\| d - \tilde{d} \right\|_1. \tag{7}$$

*2) Supervision of Shortest Paths.* We point-wisely compute $L_1$ losses between the generated and ground-truth curve points of shortest paths $\mathbf{c}_{s \to t}$ and $\tilde{\mathbf{c}}_{s \to t}$ as:

$$\ell_{\text{spath}} = \left\| \mathbf{c}_{s \to t} - \tilde{\mathbf{c}}_{s \to t} \right\|_1. \tag{8}$$

Note that since we uniformly use $M$ curve points for approximating shortest paths during training, the raw ground-truth geodesic paths typically contain different numbers of points. For the convenience of supervision, during data pre-processing, points in the raw ground-truth geodesic paths are densely interpolated and then resampled to the desired number of $M$ points.

*3) Supervision of Signed Distances.* We compute $L_1$ loss between the predicted and ground-truth signed distances $s'$ and $\tilde{s}'$ as:

$$\ell_{\text{sdist}} = \left\| s' - \tilde{s}' \right\|_1. \tag{9}$$

*4) Consistency Constraint of Curve Lengths.* For the same input pair of geodesic query points, we can simultaneously obtain their geodesic distance $d$ and shortest path $\mathbf{c}_{s \to t} = \{\mathbf{p}_m\}_{m=0}^{M-1}$. To promote the consistency between $d$ and $\mathbf{c}_{s \to t}$, we explicitly minimize the difference of curve lengths between the predicted $\mathbf{c}_{s \to t}$ and its ground-truth $\tilde{\mathbf{c}}_{s \to t} = \{\tilde{\mathbf{p}}_m\}_{m=0}^{M-1}$ as:

$$\ell_{\text{ccl}} = \left\| \sum_{m=1}^{M-1} (\|\tilde{\mathbf{p}}_m - \tilde{\mathbf{p}}_{m-1}\|_2) - \sum_{m=1}^{M-1} (\|\mathbf{p}_m - \mathbf{p}_{m-1}\|_2) \right\|_1, \tag{10}$$

where we roughly approximate the curve length as the summation of pair-wise Euclidean distances between adjacent points.

*5) Distribution Constraint of Curve Points.* Theoretically, each curve point $\mathbf{p}_m$ of the shortest path is supposed to be located on the underlying shape surface, with zero signed distance value. However, under the generative modeling framework, there is no strict guarantee to eliminate deviation between the generated geodesic curves and the corresponding shape surface. To this end, we further constrain the spatial distribution of the generated curve points by minimizing the absolute values of their signed distances, as given by:

$$\ell_{\text{dcp}} = \frac{1}{M} \sum_{m=0}^{M-1} |\mathcal{N}_\phi(\mathbf{p}_m)| \tag{11}$$

where $\mathcal{N}_\phi : \mathbb{R}^3 \to \mathbb{R}$ represents an independent neural model overfitted on the given shape for the fitting of signed distance fields in advance, whose network parameters are fixed. Given an arbitrary spatial query, $\mathcal{N}_\phi$ outputs a scalar of the corresponding signed distance value, offering a natural way of constraining the generated curve points in a differentiable manner.

## 4 Experiments

### 4.1 Implementation Details

*Testing Shapes.* We experimented with a variety of 3D shape models that are commonly adopted in the geometry processing community, covering different complexity in terms of both geometry and topology, as displayed in Figure 2. Detailed mesh statistics can be found in Table 1. The anisotropy $\tau$ quantifies the level of anisotropy in the meshes. Meshes with $\tau > 3$ are classified as anisotropic models. In our experiments, all the testing models have already been uniformly scaled into a unit sphere.

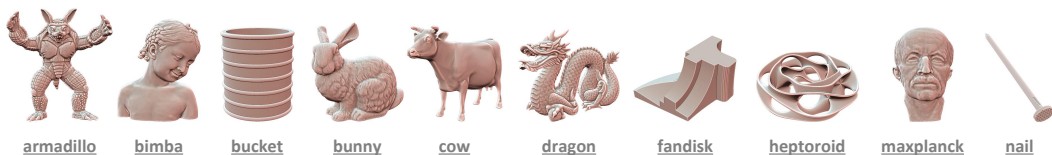

| armadillo | bimba | bucket | bunny | cow | dragon | fandisk | heptoroid | maxplanck | nail |

Figure 2: Visualization of selected testing shapes, among which the *dragon*, *bucket*, and *nail* meshes are highly anisotropic, and the *heptoroid* mesh is with high genus.

Table 1: Comparison of geodesic representation accuracy and time efficiency for SSAD querying.

| *Mesh* | *#V (K)* | $\tau$ | *Running Time (ms) of SSAD Query* | | | | *Mean Relative Error (%)* | | |
|---|---|---|---|---|---|---|---|---|---|
| | | | VTP [32] | HM [10] | fDGG [2] | NeuroGF | HM [10] | fDGG [2] | NeuroGF |
| armadillo | 173 | 1.3 | 1778 | 194 | 59 | **0.5** | 1.03 | 0.59 | **0.51** |
| bimba | 75 | 1.1 | 985 | 82 | 20 | **0.5** | 0.67 | 0.57 | **0.46** |
| bucket | 35 | 14.1 | 500 | 18 | 16 | **0.5** | 3.35 | 0.96 | **0.18** |
| bunny | 35 | 1.4 | 374 | 29 | 10 | **0.5** | 0.87 | 0.58 | **0.44** |
| cow | 46 | 1.6 | 593 | 28 | 11 | **0.5** | 2.19 | 0.57 | **0.51** |
| dragon | 436 | 12.7 | 6209 | 246 | 145 | **0.7** | 10.6 | **0.46** | 0.68 |
| fandisk | 20 | 1.4 | 359 | 14 | 4 | **0.5** | 0.88 | 0.66 | **0.35** |
| heptoroid | 287 | 2.6 | 5789 | 212 | 86 | **0.6** | 1.75 | **0.48** | 0.87 |
| maxplanck | 49 | 1.2 | 797 | 33 | 11 | **0.5** | 0.79 | 0.57 | **0.39** |
| nail | 2.4 | 4.6 | 16 | 1.4 | 0.6 | **0.4** | 2.71 | **0.42** | 0.50 |

***Training Data Preparation.*** We employed DGG-VTP [1] and fast discrete geodesic graphs (fDGG) [2] to generate the ground-truth geodesic distances and shortest paths, respectively. Specifically, we set the accuracy control parameter $\varepsilon = 10^{-7}$ for both DGG-VTP and fDGG. This parameter ensures that the computed geodesic distances and paths have accuracy comparable to the results obtained from the exact VTP method [32] using single-precision floating points. It is worth emphasizing that training NeuroGF on a given mesh does not require the utilization of geodesics between every pair of mesh vertices, which would be computationally expensive. Instead, we selected a subset of vertices and then uses the distances and paths between them. In our implementation, we sampled 20K vertices using mesh simplification tools, such as QSlim [11], for each input mesh. Our experiments show that the down-sampling strategy is sufficient for achieving satisfactory learning results. We also observed that exhaustive preparation of all-pairs geodesics for the input mesh did not further bring any obvious gain of accuracy.

***Network Configuration.*** The overall learning framework is composed of four network components. The initial query point feature embedding is achieved through a stack of three FC layers with output dimensions of $\{D_e/4, D_e/2, D_e\}$, where $D_e$ is an adjustable hyperparameter that controls the whole network size. Then comes the subsequent three learning branches for the fitting of geodesic distances, shortest paths, and signed distances, respectively. Specifically, $\mathcal{B}_{\text{gdist}}$ comprises three FC layers with output dimensions of $\{D_e/4, 64, 1\}$, $\mathcal{B}_{\text{spath}}$ is a stack of point-wise MLPs with output dimensions of $\{128, 64, 32, 3\}$, $\mathcal{B}'_{\text{sdist}}$ comprises three FC layers with output dimensions of $\{D_e/4, 64, 1\}$.

We adopted $D_e = 256$ for constructing our baseline representation model, which totally contains 259K network parameters. Note that the same number of parameters applies to all different testing shapes without changing the network structure.

***Optimization Strategy.*** We adopted the popular AdamW [20] optimizer for parameter updating with 500 training epochs, with the learning rate gradually decaying from 0.01 to 0.0001 scheduled by cosine annealing. During each training epoch, we randomly sampled around 30K spatial points as signed distance queries, 90K paired mesh vertices as geodesic distance queries, and 20K paired mesh vertices as shortest path queries, which are repeatedly consumed as inputs for 200 iterations. In the whole training phase, we specified the number of geodesic curve points as $M = 128$ and $M = 32$ for long and short shortest paths, respectively, to facilitate batch-wise processing.

***Evaluation Protocol.*** We evaluated the geodesic representation accuracy and online SSAD query efficiency of the proposed NeuroGF learning framework and made necessary quantitative comparisons with two representative computational approaches HM [10] and fDGG [2]. The quality of geodesic

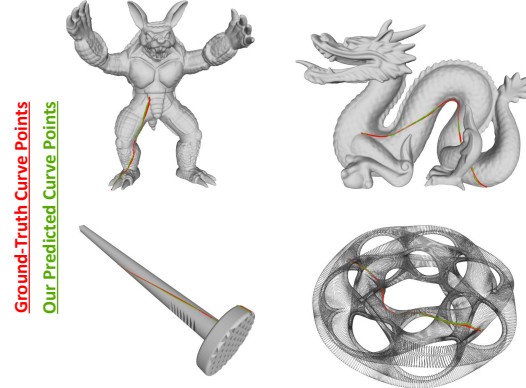

| Mesh | Chamfer-$L_1$ ($\times 10^{-2}$) |
|---|---|
| armadillo | 1.366 |
| bimba | 1.301 |
| nail | 0.354 |
| bunny | 1.559 |
| cow | 0.941 |
| dragon | 1.319 |
| fandisk | 0.822 |
| heptoroid | 2.244 |
| maxplanck | 1.434 |
| bucket | 1.183 |

Figure 3: Visual comparison of shortest paths.

Table 2: Chamfer-$L_1$ errors between ground-truth and our predicted shortest path points.

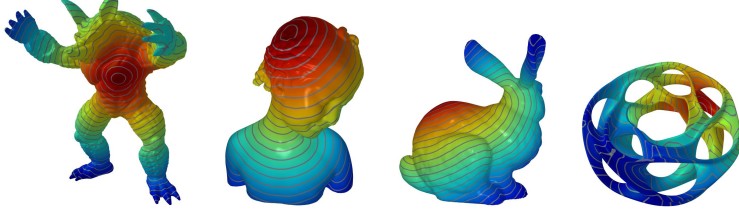

Figure 4: Visualization of our predicted geodesic distance fields. 🔍 Zoom in to see details.

distance representation is measured by the mean relative error (MRE), which can be formulated as $|d - \tilde{d}|/\tilde{d} \times 100\%$. For shortest path evaluation, we densely interpolate the predicted and ground-truth curves to around 1K points and compute their similarity as Chamfer-$L_1$ distance.

## 4.2 Quantitative Evaluations and Visualization Examples

We performed quantitative evaluations on geodesic distances produced from different approaches, as compared in Table 1, where our NeuroGF achieves the lowest mean relative errors for most testing shapes. Particularly, the performance of the heat method (HM) [10] suffers from obvious degradation when dealing with anisotropic meshes (especially for *dragon* and *bucket* with anisotropy degree $\tau > 10$), while our approach shows satisfactory robustness. We also visualized geodesic distance fields using isolines in Figure 4, which shows the smoothness of our results. We further compared the time efficiency of different approaches (including exact VTP [32] for ground truth geodesic distances and paths), where [32, 10, 2] run on the CPU (Intel i5 7500), while NeuroGF runs on the GPU (NVIDIA GeForce RTX 3090). fDGG allows the user to specify the desired accuracy of geodesic distances and paths. We set its accuracy parameter $\varepsilon = 2.5\%$ so that its results have an accuracy similar to ours, which enables us to make a fair comparison in terms of speed. Computational results show that our approach is much faster than the others for answering online SSAD queries.

In addition, the quality of the shortest paths exported from NeuroGFs is measured in Table 2 and visualized in Figure 3. We can observe that our predicted geodesic curves are highly close to ground truths even for *heptoroid*, which is characterized by highly complicated topological structures.

## 4.3 Extension to Generalizable Learning Frameworks

We made further efforts to extend the overfitting working mode of NeuroGFs introduced previously to generalizable learning frameworks. More specifically, as illustrated in Figure 5, we designed three versions of generalizable NeuroGFs using (a) *autodecoder-based* (i.e., DeepSDF-like [29]), (b) *point transformer-based*, and (c) *graph convolution-based* feature extraction strategies. Notably, NeuroGF equipped with (a) or (b) for shape encoding can directly work on point clouds during testing. We used the popular ShapeNet [7] mesh dataset pre-processed by [42], covering 13 different shape categories. We collected 3000 models from 8 categories as our training set. For each model, we only sparsely generated 2K ground-truth training pairs. For evaluation, we constructed different testing sets: 1)

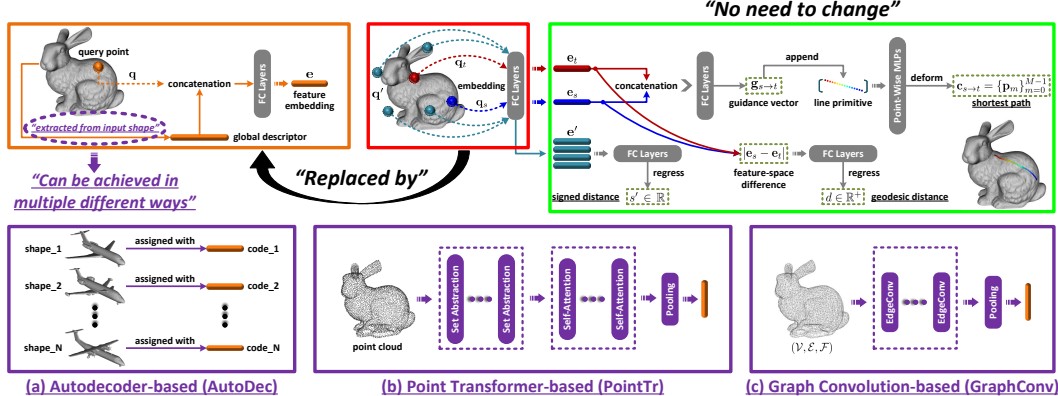

Figure 5: Illustration of extending our NeuroGFs to generalizable learning frameworks. From the architectural point of view, we only need to replace the single-object query point embedding module **(red box)** by a multi-object embedding module **(orange box)**.

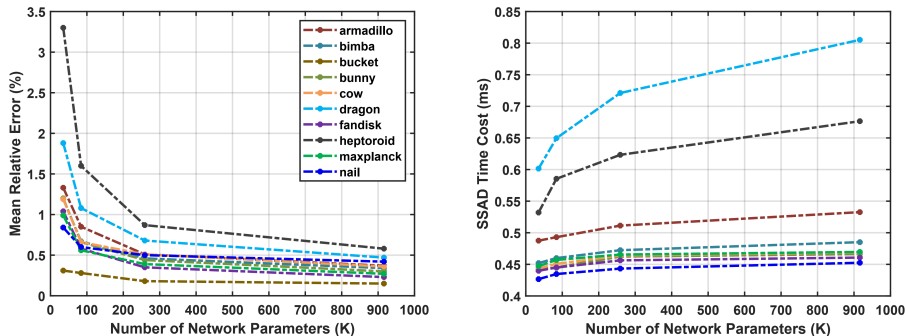

Figure 6: Statistics of SSAD geodesic distance querying with different network complexity.

SN-Airplane, SN-Chair, and SN-Car are collected within the same categories of *airplane*, *chair*, and *car*, each of which containing 500 models; 2) SN-8x50 is collected from the same 8 categories as in the training set, but each shape is unseen during training. 3) SN-5x50 is collected from the other 5 different categories.

Quantitative results are reported in Table 3. The testing results on (a) validate the category-specific representation capability (with about 3% MRE). The testing results on (b) show that our extended approach equipped with a powerful deep point encoder works well on point clouds for both seen and unseen categories. (c) further incorporates mesh connectivity cues, thus achieving better performance.

### 4.4 Ablation Studies

In the preceding experiments, we uniformly configured $D_e = 256$, corresponding to 259K network parameters. Here, we further investigated the effects of scaling the network size by changing $D_e$ to 64, 128, and 512, leading to three different variants with 35K and 84K, and 916K parameters, respectively. As illustrated in Figure 6, as the network complexity increases, the geodesic representation accuracy gets stably enhanced. Furthermore, to reveal the specific influences of the different learning components and supervision objectives involved in our approach, we performed necessary ablative analyses as presented in Table 4. We can observe that removing any of them leads to some degree of performance degradation, which demonstrates their necessity and effectiveness.

## 5 Conclusion and Discussion

This paper made the first effort to investigate learning-based neural implicit representations for 3D surface geodesics in both overfitted and generalizable working modes. The proposed NeuroGF achieves accurate and highly efficient point-to-point queries of both geodesic distances and shortest paths, which can be naturally combined with the learning of signed distances to further produce a

Table 3: MRE (%) performances of generalizable NeuroGF learning frameworks equipped with different global shape feature extractors, including: (a) AutoDec, (b) PointTr, and (c) GraphConv.

| (a) | SN-Airplane | SN-Chair | SN-Car | (b) | SN-8x50 | SN-5x50 | (c) | SN-8x50 | SN-5x50 |
|---|---|---|---|---|---|---|---|---|---|
| *AutoDec* | 3.03 | 3.91 | 2.78 | *PointTr* | 3.28 | 4.16 | *GraphConv* | 2.94 | 3.55 |

Table 4: Influences of different learning components and supervision objectives, where the results are averaged on all testing shapes. The right two columns show the representation accuracy of our predicted geodesic distances and shortest paths (the lower, the better). The averaged statistics of our full implementation in terms of the two metrics are $0.49\%$ and $1.25 \times 10^{-2}$. In particular, we mark the relative change within each bracket to facilitate comparison.

| $\mathcal{B}_{\text{gdist}}$ | $\mathcal{B}_{\text{spath}}$ | $\mathcal{B}'_{\text{sdist}}$ | $\ell_{\text{ccl}}$ | $\ell_{\text{dcp}}$ | Mean Relative Error (%) | Chamfer-$L_1$ ($\times 10^{-2}$) |
|---|---|---|---|---|---|---|
| ✗ | | | | | - | 1.46 (↑ 0.21) |
| | ✗ | | | | 0.61 (↑ 0.12) | - |
| | | ✗ | | | 0.57 (↑ 0.08) | 1.31 (↑ 0.06) |
| | | | ✗ | | 0.52 (↑ 0.03) | 1.35 (↑ 0.10) |
| | | | | ✗ | 0.50 (↑ 0.01) | 1.38 (↑ 0.13) |

unified geodesic and geometric representation structure. Experiments demonstrated that our approach achieves comparable, if not better, geodesic representation accuracy against previous representative computational and optimization-based approaches. We believe that NeuroGF opens up new and promising paradigms for the complicated problem of geodesic representation and computation.

Table 5: Chamfer-$L_1$ ($\times 10^{-2}$) errors between ground-truth and our predicted shortest path points after post-processing.

| armadillo | bimba | nail | bunny | cow | dragon | fandisk | heptoroid | maxplanck | bucket | average |
|---|---|---|---|---|---|---|---|---|---|---|
| 1.131 | 1.126 | 0.347 | 1.325 | 0.804 | 1.070 | 0.766 | 1.994 | 1.248 | 1.079 | **1.09** |

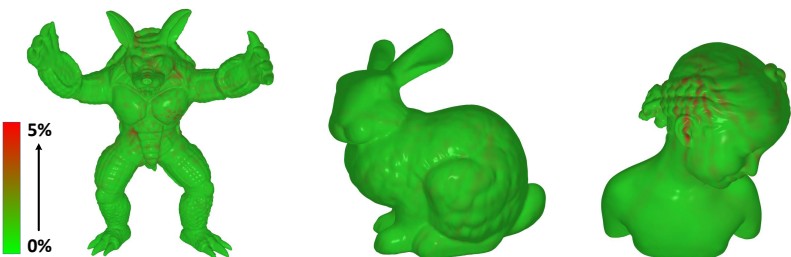

Figure 7: Error distribution of our predicted geodesic distance fields. 🔍 Zoom in to see details.

While we have demonstrated superior runtime performance of NeuroGF in answering point-to-point distance queries and solving the SSAD problem, our approach is still subject to a few limitations. There is no guarantee that the generated geodesic path strictly lies on the underlying surface, as the path points may exhibit varying degrees of deviation from the surface. To enhance the accuracy of geodesic paths, post-projection/refinement techniques can be employed, utilizing direction cues from signed distance fields. As shown in Table 5, the shortest path representation accuracy will further improve after pre-processing for making curve points locate exactly on the underlying surface. Although our approach achieves highly stable and satisfactory mean relative errors, it is worth noting that the metric of maximal errors can sometimes reach higher values, as depicted in Figure 7, where the relative error can rise up to approximately 5%. To address this, incorporating more advanced optimization strategies, such as online hard example mining, may be necessary.

## Acknowledgement

This work was supported by the Hong Kong Research Grants Council under Grants CityU 11202320 and CityU 11219422.

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

# A  Appendix

*1) Visualization of MSAD Geodesic Distance Fields.* We randomly selected 5 source points on the shape surface to produce the resulting MSAD geodesic distance field, as illustrated in Figure A1. Besides, we also visualized the triangulation of the testing meshes in Figure A2.

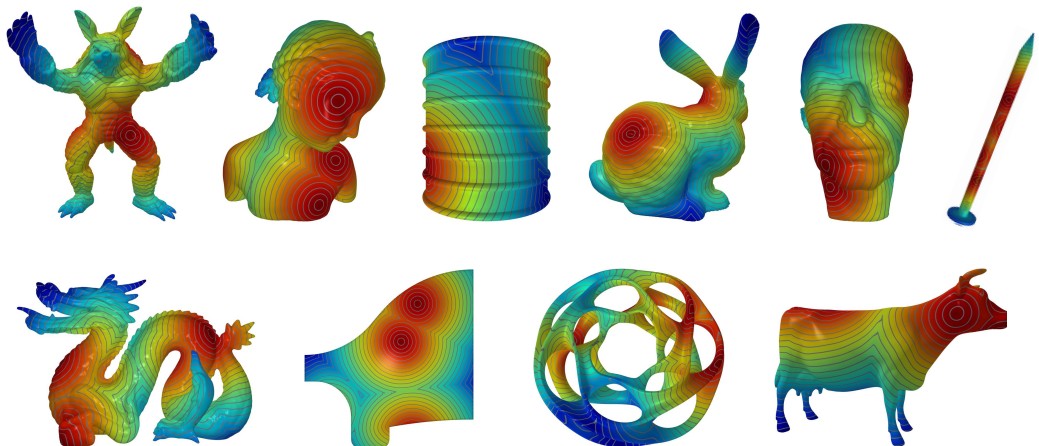

Figure A1: Visualization of MSAD geodesic distance fields computed by our approach. 🔍 Zoom in to see details.

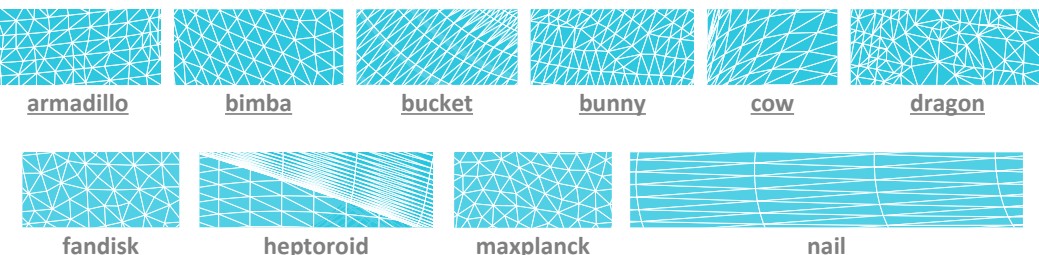

Figure A2: Visualization of mesh triangulations.

*2) Visualization of SSAD Geodesic Distance Fields.* We provided more visual comparisons of the SSAD geodesic distance fields exported from different approaches, as presented in Figure A3. To facilitate comparisons, we also marked the quantitative metrics of geodesic representation accuracy below each example computed from different approaches. In particular, Figure A4 presents close-up views for regions on the anisotropic *nail* model.

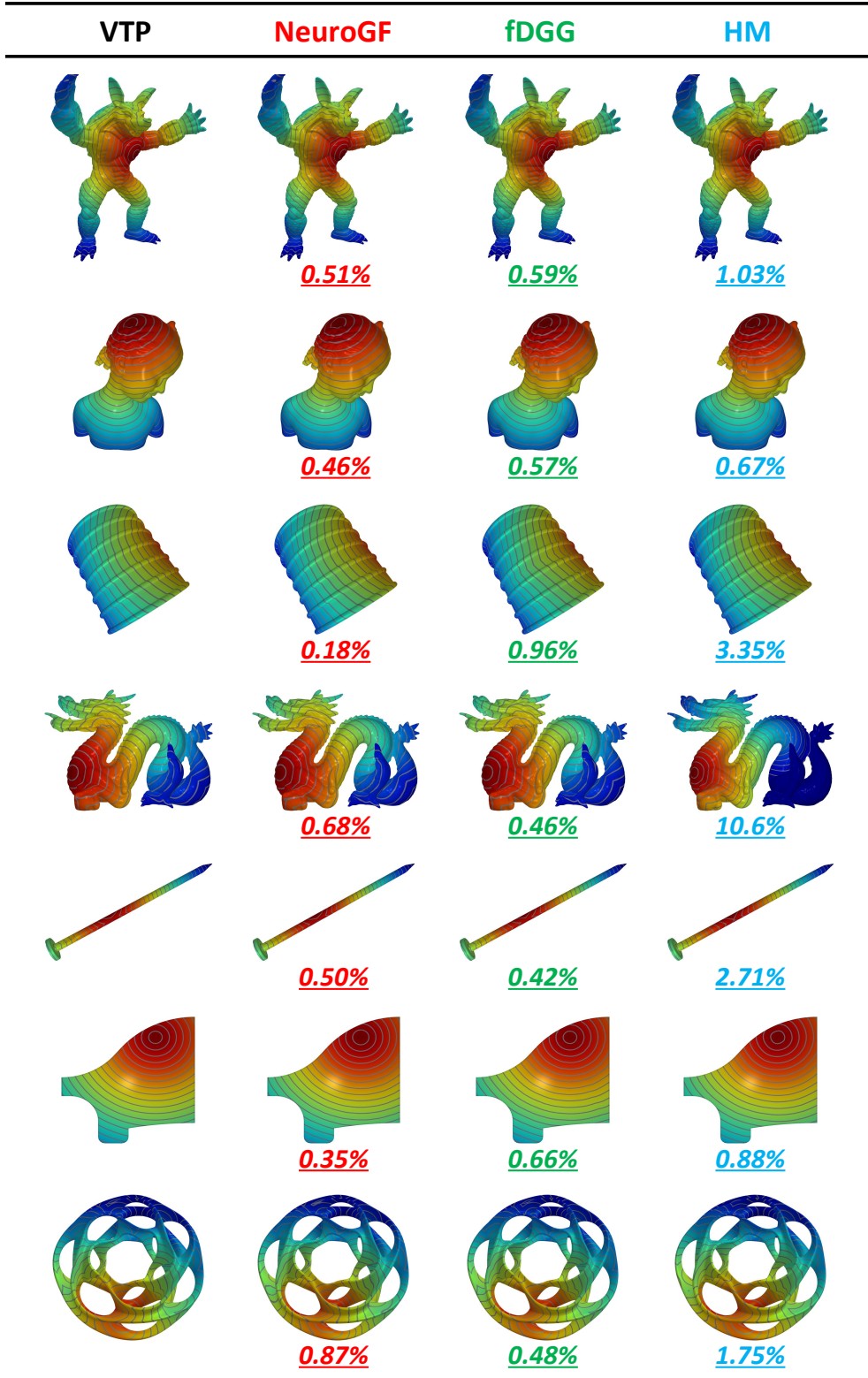

Figure A3: Visual comparison of SSAD geodesic distances. Below each example, we also marked the corresponding mean relative error (%). 🔍 Zoom in to see details.

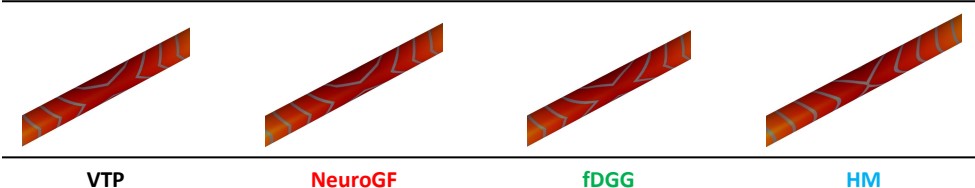

| VTP | NeuroGF | fDGG | HM |

Figure A4: Close-up views for geodesic distance fields on the anisotropic *nail* model.

***3) Offline Training with Limited Ground-Truth Geodesics***. For the preprocessing stage of preparing the required ground-truth geodesics for offline training of NeuroGFs, during which 4 data generation scripts run in parallel, we explored the effects of using different numbers of source points, i.e., 10000, 1024, 512, 256, 128, and 64. For each source point, we only preserved its geodesic distances between 4096 target points and its shortest paths between 2048 target points. As compared in Table A1, our approach can still achieve relatively satisfactory geodesic representation performances when only a limited amount of source points are exploited to produce ground-truth training data. Besides, we can also observe that using a much larger amount of training samples (i.e., with 10000 source points) only leads to insignificant performance gains.

Furthermore, note that here our preprocessing stage is not well-optimized, since we need to do lots of disk writing operations. We can directly integrate the approaches used for data preparations (fDGG [2] and DGG-VTP [1]) to Python to avoid costly disk writing time in the future. Comparatively, the recent work SEP [12] requires around 30 minutes for preprocessing a mesh with 20K vertices. Its time complexity of preprocessing is $O(f(n)+m^3n)\sqrt{n})$, where $n$ is the number of mesh vertices and $f(n)$ is the time complexity of the used SSAD algorithm, which is at least $O(n)$ time for fDGG [2]. We can see that this time complexity increases more than linearly in the number of mesh vertices. Thus, we can estimate that for the *dragon* mesh model with more than 400K vertices, SEP would require more than 10 hours to finish preprocessing. This is much longer than our preprocessing stage using 1024 source points, which achieves geodesic representation accuracy comparable to SEP.

In addition, we also evaluated training with highly sparse ground-truth pairs. As shown in Table A2, when training NeuroGFs with only 8K and 2K ground-truth pairs, the errors do not rise up drastically, still maintaining relatively satisfactory.

Table A1: Effects of offline training with different amounts of paired ground-truth geodesics on the *dragon* model. In the right two columns, we respectively record the preprocessing (Prep.) time cost for generating ground-truth geodesic distance (G.D.) and shortest path (S.P.) data.

| #Sources | MRE (%) | Chamfer-$L_1$ ($\times 10^{-2}$) | Prep. G.D. (minutes) | Prep. S.P. (hours) |
|---|---|---|---|---|
| 10000 | 0.66 | 1.297 | 119.1 | 21.6 |
| 1024 | 0.68 | 1.319 | 12.2 | 2.2 |
| 512 | 0.75 | 1.397 | 6.1 | 1.1 |
| 256 | 0.83 | 1.580 | 3.0 | 0.6 |
| 128 | 1.09 | 2.044 | 1.5 | 0.3 |
| 64 | 1.63 | 2.926 | 0.8 | 0.14 |
| 32 | 2.21 | 3.725 | 0.4 | 0.07 |

Table A2: Effects of training NeuroGFs with highly sparse ground-truth pairs.

| #GT-Pairs | MRE (%) | #GT-Pairs | MRE (%) | #GT-Pairs | MRE (%) |
|---|---|---|---|---|---|
| Original | 0.53 | 8K | 4.05 | 2K | 5.62 |

***4) Memory Footprint of Online Point-to-Point Geodesic Query***. We compared the memory footprints of different approaches for online answering the geodesic distance between an input pair of source and target points. The quantitative results are reported in Table A3, where we can observe that our approach consistently maintains satisfactory memory efficiency. Thanks to our implicit querying paradigm, the resulting memory footprint is independent of the complexity of the given shape. Instead, for both the two competing approaches of HM [10] and fDGG [2], their memory footprints increase when dealing with larger meshes.

Table A3: Comparison of memory efficiency for online point-to-point geodesic query.

| Mesh | #V (K) | Memory Footprint (MB) of Point-to-Point Geodesic Query | | |
|---|---|---|---|---|
| | | HM [10] (on CPUs) | fDGG [2] (on CPUs) | NeuroGF (on GPUs) |
| armadillo | 173 | 272 | 60 | 10 |
| bimba | 75 | 138 | 26 | 10 |
| bucket | 35 | 45 | 15 | 10 |
| bunny | 35 | 56 | 13 | 10 |
| cow | 46 | 70 | 17 | 10 |
| dragon | 436 | 597 | 152 | 10 |
| fandisk | 20 | 29 | 7 | 10 |
| heptoroid | 287 | 690 | 108 | 10 |
| maxplanck | 49 | 85 | 17 | 10 |
| nail | 2.4 | 2.9 | 1.3 | 10 |

**5) *Evaluation on Large-Scale and Real-World Meshes*.** We experimented with a much denser version of the dragon model with 1.5M vertices and another classic graphics model lucy with 6.9M vertices. As reported in Table A4, our approach works well on these two highly dense mesh models. As illustrated in Figure A5, in addition to the object models used in the paper for testing, here we further selected several representative large-scale real-world mesh models, covering (a) indoor room, (b) urban scene, and (c) street view, which are from diverse dataset sources. It can be observed that our approach still achieves satisfactory performances on the three scene-level meshes with MRE rates of 0.37%, 1.14%, and 0.23%, respectively.

Table A4: MRE (%) performances for a much denser version of the *dragon* model and another popular *lucy* model.

| Mesh | #V (M) | MRE (%) | Chamfer-$L_1$ ($\times 10^{-2}$) |
|---|---|---|---|
| dragon | 1.5 | 0.70 | 1.335 |
| lucy | 6.9 | 0.58 | 1.404 |

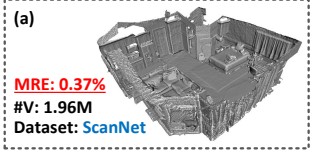 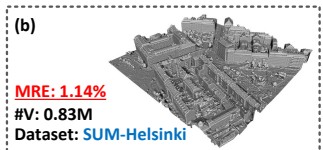 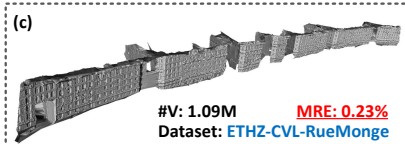

(a) MRE: 0.37% #V: 1.96M Dataset: ScanNet
(b) MRE: 1.14% #V: 0.83M Dataset: SUM-Helsinki
(c) #V: 1.09M MRE: 0.23% Dataset: ETHZ-CVL-RueMonge

Figure A5: Experimental evaluations on large-scale real-world meshes (with up to 2 million vertices).

**6) *Geometry Representation Recovery*.** As pointed out in the paper, NeuroGFs jointly encode both 3D geometry and geodesics in a unified neural representation structure, although geodesic information is our major focus and geometric information just serves as a byproduct in this work.

Here, we provided necessary qualitative and quantitative evaluations for signed distance field information encoded in NeuroGFs. We deduced the predictions of signed distance values on a uniformly-distributed 3D grid with the resolution of $512^3$, and then measured the $L_1$ differences between the predicted and ground-truth signed distances, as reported in Table A5. For visualization, we applied the Marching Cubes [48] algorithm for isosurface extraction. The resulting mesh reconstructions are displayed in Figure A6.

**7) *Time Cost of Per-Model Overfitting*.** Practically, since NeuroGF is an offline per-input overfitting process, users can flexibly control the trade-off between training time and fitting accuracy. Typically, the training time cost for achieving comparable performances as reported in the paper is "minute-level". Table A6 provides statistics averaged on all the 10 testing models, showing that the convergence speed is fast. For example, it only takes less than half a minute to reach the MRE lower than 3%. From our experience, in most cases, the whole training effects will saturate within 10 minutes. Further extending the training process can only bring marginal performance gains at a slow pace.

Table A5: Mean $L_1$ errors between our predicted and ground-truth signed distances.

| Mesh | armadillo | bimba | bucket | bunny | cow | dragon | fandisk | heptoroid | maxplanck | nail |
|---|---|---|---|---|---|---|---|---|---|---|
| $L_1$ $(\times 10^{-3})$ | 1.22 | 0.97 | 0.68 | 0.97 | 0.86 | 1.28 | 0.67 | 1.01 | 1.03 | 0.45 |

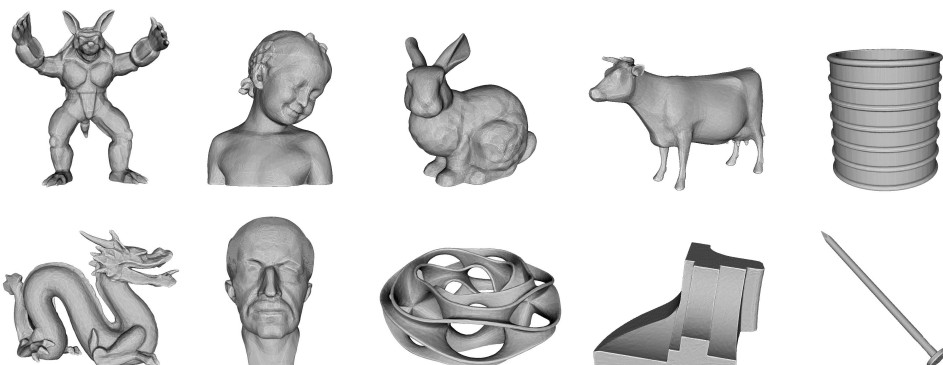

Figure A6: Visualization of mesh reconstruction deduced from NeuroGFs.

Table A6: Time cost for overfitting a single input 3D model.

| MRE | <3% | <2% | <1% |
|---|---|---|---|
| Training Time Cost | 0.4min | 1.7min | 7.9min |

Table A7: Ablation studies on different variants of our technical implementations, where "variant (1)" means adding position encoding before fed into FC layers for query point feature embedding, "variant (2)" means replacing $L_1$ loss with $L_2$ loss for supervisions, and "variant (3)" means computing Chamfer distance between generated and ground-truth curve points of shortest paths $\mathbf{c}_{s \to t}$ and $\tilde{\mathbf{c}}_{s \to t}$ for the formulation of $\ell_{\text{spath}}$ (Eq. (8) in the paper).

| Implementation Variants | dragon | | heptoroid | |
|---|---|---|---|---|
| | MRE | Chamfer-$L_1$ | MRE | Chamfer-$L_1$ |
| (0) Original Implementation | 0.68 | 1.319 | 0.87 | 2.244 |
| (1) Adding PosEnc | 0.62 | 1.253 | 0.79 | 1.921 |
| (2) $L_2$ Supervisions | 0.71 | 1.345 | 0.88 | 2.227 |
| (3) Chamfer Loss for $\ell_{\text{spath}}$ | 0.66 | 1.286 | 0.82 | 2.149 |

*8) More Ablation Studies*. We conducted more ablation studies in Table A7. We applied position encoding to input query point coordinates before high-dimensional feature embedding. We can observe that the resulting performances on the two complex meshes *dragon* and *heptoroid* are improved in different degrees, demonstrating the potential of exploring more advanced position encoding strategies. We also explored different ways of formulating the learning objectives in terms of the choice of distance metric. For example, we can choose to use $L_2$ loss instead of $L_1$ to calculate all learning objectives. Besides, for the formulation of shortest path supervision (i.e., formulated as Eq. (8) in the paper), we can also choose the popular Chamfer distance to supervise the curve deformation process. We can observe that in most cases using $L_2$ loss causes different degrees of performance degradation, and Chamfer distance leads to further performance improvement. Still, in our implementation, we did not adopt Chamfer distance as our final choice because of its additional computational burden and memory cost.

