# OpenReview forum: "NeuroGF: A Neural Representation for Fast Geodesic Distance and Path Queries"
_NeurIPS.cc/2023/Conference — NeurIPS 2023 poster_

### Official Review · Reviewer_Ujtd · 2023-07-03

**Soundness:** 3 good
**Presentation:** 4 excellent
**Contribution:** 2 fair
**Rating:** 4
**Confidence:** 5

**Summary:**

The paper proposes a neural implicit representation of a 3d surface that enables querying for geodesic lengths and paths between points on the surface. A neural network is overfitted to one given surface. The input query points are embedded into high dimensional features whose euclidean distance represents the desired geodesic distance. Another branch learns to receive the high dimensional features to deform a straight line to the geodesic between the two points. A final branch learns to represent the SDF of the surface. The network is trained via strong supervision w.r.t given geodesics computed on the mesh. The results are shown to produce high accuracy when overfitted to well-known graphics models.

**Strengths:**

The idea of overfitting to represent a geodesic paths can be quite useful, especially considering the ability to parallelize many queries through the network as one batch.

**Weaknesses:**

I find the work to provide a good technical solution to the given problem, however I cannot champion the paper: The technical contribution is quite limited. Beyond the core idea of overfitting to geodesics of a given shape, the rest of the approach is quite straightforward. As such this feels more as an application than a NeurIPS paper.

**Questions:**

1. Why use the SDF at all? why not provide the first 2 branches + the original mesh as the full representation? if the SDF is completely decoupled, what good does it bring.
2. is there any guaranteed that the geodesic paths actually lie on the isosurface of the SDF? if they are not, how is the difference reconciled?

**Limitations:**

Yes

---

> ### Author Rebuttal · Authors · 2023-08-10
>
> ### **[Rebuttal to Reviewer Ujtd]**
>
> ### **[W1]** *Limited technical contribution due to the quite straightforward approach.*
>
> **Response:** Thanks for your recognition of the usefulness of our core idea and technical solution. **Still, we beg to differ with the judgment that our work shows limited contribution only because our technical implementations are quite straightforward.**
>
> On one hand, as the very first attempt to adapt the neural implicit modeling paradigm for geodesics representation, we resort to a concise yet highly effective learning framework to verify the potential of such a completely new geodesics representation paradigm.
>
> On the other hand, the formulation of the shortest path as a discrete sequence of ordered curve points requires an appropriately designed learning structure, and our corresponding adaptation of previous "folding-style" 3D shape reconstruction decoders [14, 43] to a 1D version for curve deformation also has its technical value and thus is not quite straightforward as the reviewer commented.
>
> Besides, we also refer the reviewer to our extensions of generalizable NeuroGFs (asked by Reviewer yBGS), as given in Figure R1 and Table R4 of the uploaded PDF file as well as **our responses to Reviewer yBGS [W1]**. These explorations are also of significance.
>
> In general, we believe that this work does bring novel insights and meaningful technical contributions for the problem of geodesics representation and computation and thus qualified for the NeurIPS research community.
>
> ### **[Q1]** *Necessity of the SDF learning branch.*
>
> **Response:** There seems to be a **misunderstanding** of our approach. Maybe our descriptions in lines 180-185 of the paper is not clear enough. In fact, during the testing phase, we can say that the SDF learning branch is completely decoupled. However, during the training phase, since the query points for both SDF and geodesics share the same FC layers for feature embedding, the learning process of the SDF branch will have an impact on the other branches. And in our ablation studies (Table 3 of the paper), the effectiveness of the SDF learning branch has been validated (i.e., removing the SDF learning branch causes performance degradation of geodesics). Moreover, as discussed in **our response to Reviewer yBGS [W5]**, the learning of SDF also benefits the convergence speed of the geodesics learning process.
>
> ### **[Q2]** *Difference between the geodesic paths and the iso-surface of the SDF.*
>
> **Response:** As both quantitatively and qualitatively illustrated in the paper, the generated shortest paths are close enough to the underlying iso-surface, meaning that we can conveniently deduce post-processed shortest paths whose curve points are exactly located on the surface by straightforward local projection. Here, we simply implement this process by locally sampling surface points and then performing nearest-neighbor matching for the raw outputs of curve points. As shown in Table R2 of the uploaded PDF file, **such a post-processing procedure consistently brings further accuracy improvement for the prediction of shortest paths on ALL the testing meshes**. On average, the error decreases from the original $1.25 \times 10^{-2}$ to $1.09 \times 10^{-2}$.

---

> > ### Author Response · Authors · 2023-08-21
> > **Looking forward to receiving your feedback**
> >
> > Dear **Reviewer Ujtd**
> >
> > Thank you for taking the time to review our submission. As the discussion phase between the reviewers and authors is coming to the end, may we know whether there are still unsolved concerns from you? We are pleased to address them. Looking forward to receiving your feedback.
> >
> > Best regards,
> >
> > The authors

---

### Official Review · Reviewer_nwTQ · 2023-07-04

**Soundness:** 3 good
**Presentation:** 3 good
**Contribution:** 3 good
**Rating:** 6
**Confidence:** 3

**Summary:**

The authors propose a solution to the estimation of the geodesic distance between a pair of points (source, target) on a mesh. The proposed approach relies on an implicit field. In particular, the implicit field learns/memorizes the distance between each pair of points. The authors sample a subset of mesh vertices and pre-compute the geodesic distance between them. These points are later used to optimize the network weights (implicit field). Each point is converted into a feature vector by a shared MLP. The difference between source and target features is used to predict the geodesic distance between the two points. On top of this, a second branch predicts the implicit field representing the shape itself (using the shared MLP features as input). Finally, a third branch using the same features, as concatenation, predicts the geodesic path connecting the source to the target as a set of points.

The authors demonstrate the effectiveness of the paper on a handful of meshes, showing numerical and qualitative results, while comparing with other approaches.

**Strengths:**

The paper addresses an important problem. The manuscript describes the idea intuitively, clearly, and in a well-structured manner. The solution is interesting, original, and well-crafted. It leverages past approaches to achieve a different goal. Furthermore, the method is incredibly faster than chosen baselines while being accurate - both in terms of geodesic distance, geodesic path, and reconstruction error. This is possible with the use of NN, as it requires a single forward pass, compared to classic methods which require extensive computations.

**Weaknesses:**

The proposed method uses a shared MLP to predict a set of features from 3D points. However, the input is not limited to points belonging to the surface. The same is the output of the geodesic path (one of the branches).
This seems to be quite an important limitation which may severely prevent the application of this method. Indeed, the authors introduce an extra loss term 5) to "force" the points on the curve to belong to the surface. Yet there is no guarantee this happens.
Furthermore, to apply this technique we have to optimize a neural network, which takes time while the classic methods do not require such expensive pre-processing.


**Questions:**

 The solution is interesting and leaves me with some questions, it would be great if the authors can address some of these:
- how much time does the pre-processing time require? ie optimizing the network + sampling points + estimating distances for training
- the current formulation for the path relies on AtlasNet/Foldingnet, have the authors thought about using a different approach that cannot predict points off the surface? For example, the use of a transformer that predicts the index of the next point on the path.
- how well does the method scale with very large meshes? (#V  > 1M)
- how robust is the method on sparse meshes where very few vertices can be used to define geodesic distances? For example, on a cube where the only vertices are at the corners, and the query points are pairs on the faces (assuming this is possible).

I thank the authors for the time spent addressing any of these concerns.

**Limitations:**

There are no negative social impacts of this work.
The authors discussed the limitation of this work, although they did not detail properly the time requirement of the method (such as pre-processing time). I would recommend the authors add these details either in the main paper or in the supplementary. Furthermore, it would be great if the authors could evaluate the approach on a very diverse set of meshes, such as a cube and a super-dense version of "dragon".

---

> ### Author Rebuttal · Authors · 2023-08-10
>
> ### **[Rebuttal to Reviewer nwTQ]**
>
> ### **[W1]** *1) The input is not limited to points belonging to the surface; 2) The output geodesic path is not guaranteed to lie on the surface; 3) Have to optimize a neural network for each new input.*
>
> **Response:** For the first issue, we argue that it is not an influencing factor in practice, because users will naturally input paired query points that are either vertices of the target mesh model or sampled from the mesh surface. Thus, it would be unnecessary to impose restrictions on the input geodesic query points.
>
> For the second issue, as both quantitatively and qualitatively illustrated in the paper, the generated shortest paths are close enough to the underlying iso-surface, meaning that we can conveniently deduce post-processed shortest paths whose curve points are exactly located on the surface by straightforward local projection. Here, we simply implement this process by locally sampling surface points and then performing nearest-neighbor matching for the raw outputs of curve points. As shown in Table R2 of the uploaded PDF file, such a post-processing procedure consistently brings further accuracy improvement for the prediction of shortest paths on ALL the testing meshes. On average, the error decreases from the original $1.25 \times 10^{-2}$ to $1.09 \times 10^{-2}$.
>
> For the third issue, as can be found in our below response to your question [Q1], although our approach is an overfitting process, its convergence speed is fast. Moreover, in terms of the overfitting paradigm, NeuroGF inherits from previous works like SEP [13] (C. Gotsman, et al., SIGGRAPH Asia-22'), which can be viewed as the process of "compressing geodesics". As summarized in [13], this setting well suits many interactive applications where rapid computation of arbitrary point-to-point geodesics on very large meshes is required.
>
> ### **[Q1]** *Time cost of the whole pre-processing process (including network training).*
>
> **Response:** The time cost statistics of the data pre-processing procedures (not including network training) have already been provided in Table S1 of the supplementary material, and detailedly discussed on page 3, lines 15-33 of the supplementary material.
>
> As for the network training, practically, since NeuroGF is an offline overfitting process, users can flexibly control the trade-off between training time and fitting accuracy. Typically, the training time cost for achieving comparable performances as reported in the paper is "minute-level". Below we provide statistics averaged on all the 10 testing models, showing that the convergence speed is fast. For example, it only takes less than half a minute to reach the MRE lower than 3\%.
>
> | MRE | <3\% | <2\% | <1\% |
> | :----: | :----: | :----: | :----: |
> | **Training Time** | 0.4min | 1.7min | 7.9min |
>
> From our experience, in most cases, the whole training effects will saturate within 10 minutes. Further extending the training process can only bring marginal performance gains at a slow pace.
>
>
> ### **[Q2]** *Think about different approaches that cannot predict path points off the surface.*
>
> **Response:** Thanks for your valuable comments and insightful consideration. For the current learning framework implemented in a generative fashion, it is basically impossible to ensure that all generated points exactly lie on the surface. We agree that predicting the next point along the path might be a viable scheme worthy of further exploring. Yet, considering the limited time for rebuttal and the complexity of designing a completely new framework, we cannot conduct systematic explorations for this issue. We may leave it as a promising direction for future study.
>
> ### **[Q3]** *How well does the method scale with very large meshes (\#V > 1M).*
>
> **Response:** Following your instructions, we added more diverse experiments on very large-scale meshes. Here, we experimented with a much denser version of the *dragon* model with 1.5M vertices and another classic graphics model *lucy* with 6.9M vertices. As reported in the Table below, our approach works well on these two highly dense mesh models.
>
> | Mesh | \#V (M) | MRE (\%) | Chamfer-$L_1$ ($\times 10^{-2}$) |
> | :----: | :----: | :----: | :----: |
> | dragon | 1.5 | 0.70 | 1.335 |
> | lucy | 6.9 | 0.58 | 1.404 |
>
> Moreover, we further performed evaluations on three large-scale real-scanned meshes covering (a) indoor room, (b) urban scene, and (c) street view, which are from diverse dataset sources. As shown in Figure R2 of the uploaded PDF file, our approach still achieves satisfactory performances (with MRE rates of 0.37\%, 1.14\%, and 0.23\%, respectively) on the three scene-level meshes with million-scale amounts of vertices.
>
>
> ### **[Q4]** *Robustness to extremely sparse meshes with very few vertices (e.g., a cube model with only vertices at corners).*
>
> **Response:** As you suggested, here we add an experiment with a cube model, with only 8 vertices at corners and 12 triangles. The total number of available ground-truth pairs is 28. The testing data are prepared by performing decimation on the cube to deduce a much denser mesh, on which we calculated enough pairs of geodesic data for performance evaluation. It turns out that our approach fails under this extreme case, achieving about 70\% MRE rate.
>
> Yet, we argue that in practice we can conveniently perform necessary mesh decimation to enable preparing more ground-truth pairs. For example, when we decimate the cube model to about only 100 vertices for ground-truth preparation, the resulting performance drastically improves to 5.24\% MRE.
>
> In addition to this extreme case, we would also refer the reviewer to our response to Reviewer oZNk [W2] for our results when the ground-truth data are sparse.

---

> > ### Comment · Reviewer_nwTQ · 2023-08-13
> >
> > Thank you for taking the time respond to my concern in well a structered manner and the additional evaluation.
> >
> > ### W1
> > As the shortest path may not belong to the mesh, you are suggesting to perform a local projection. This may solve the problem, or introduce artifacts in concave regions. Either way, the numerical results suggest the error (10^-2) is relatively high compared to what I would agree to be accurate predictions.
> >
> >
> > ### Q1
> > Thank you for the comprehensive response. Overall a single query may take a long time, even hours, (preprocessing of 21 hours from Table S1 + 119 minutes), this time being amortized with multiple queries. While for a mesh-based approach, each query is computed at runtime. A plot showing #query vs total training time would show the benefits of this approach.
> > I struggle to see where this approach (for SSAD or MSAD) would be applicable, can authors cite any applications where a 10^-2 error in the geodesic error would be acceptable?
> >
> >
> > ### Q3
> > Thank you for the additional evaluation.
> >
> > ### Q4
> > Thank you for your response and additional evaluation. I understand with sparse GT samples the accuracy degrades, this is peraphs a limitation worth mentioning in the manuscript.

---

> > > ### Author Response · Authors · 2023-08-15
> > >
> > > **Response:** We sincerely appreciate your evaluation of our rebuttal materials and the active discussion. Here, we further respond to your raised questions with more targeted explanations.
> > >
> > >
> > > In terms of downstream application scenarios, it is worth clarifying that the geodesic representation accuracy achieved by our approach is adequate for supporting a rich variety of geometry processing and modeling tasks. Note that the popular classical geodesic algorithms (e.g., *heat method* (HM) and *fast marching method* (FMM)) typically produce results with higher than 1\% errors (sometimes even much higher for anisotropic meshes). Hence, to answer your question about "*applications where a $10^{-2}$ error in the geodesic error would be acceptable*", basically many papers where (original implementations or variations of) HM and FMM are used to produce geodesic information can be taken as examples. As listed in the following, [A, B, C] involve HM-based and FMM-based approaches for geodesic computation, and [D] simply resorts to constructing k-NN graphs followed by the Floyd’s shortest path algorithm, which leads to even higher errors. Also, it can be used for supporting various interactive geometry processing and modeling tasks, such as texture transfer, decal placement, remeshing, smoothing, splines on surfaces, as reported in [E], in which the authors adopted a low-dimensional Euclidean embedding for computing geodesics with approximation error around 5\%.
> > >
> > >
> > > -- [A] A. Poulenard, et al., "Multi-Directional Geodesic Neural Networks via Equivariant Convolution," in ACM TOG, 2018.
> > >
> > >
> > > -- [B] B. L. Bhatnagar, et al., "Multi-Garment Net: Learning to Dress 3D People from Images," in Proc. ICCV, 2019.
> > >
> > >
> > > -- [C] R. Wiersma, et al., "CNNs on Surfaces using Rotation-Equivariant Features," in ACM TOG, 2020.
> > >
> > >
> > > -- [D] Z. Li, et al., "Geodesic Self-Attention for 3D Point Clouds," in Proc. NeurIPS, 2022.
> > >
> > >
> > > -- [E] D. Panozzo, et al., "Weighted Averages on Surfaces," in ACM TOG, 2013.
> > >
> > >
> > > Moreover, we would like to further explain the actual application scenarios of our approach in an overfitting paradigm. To facilitate the intuitive understanding of what we say "interactive applications with rapid and extensive geodesics querying", here we illustrate a very specific case. In the practical industrial scenarios for game/movie/animation, many important 3D asset models such as characters, animals, or other types of objects, need to be repeatedly manipulated, with extensive querying of geodesics at each time. In this case, it can be highly inefficient to apply conventional computational algorithms for many times, and it is even more impractical to store all pre-computed geodesics for every pair of source-target points due to extremely huge memory cost. For this situation, our NeuroGFs can serve as a good choice. Intuitively, once the offline training (i.e., overfitting) process is completed, the neural network model can be viewed as "an attribute" of the target mesh model, just like other attributes like texture. This trained neural network model is quite compact (around 1MB), fast to online query, and can be permanently stored together with the original mesh model as "a compression of its complete geodesics information".
> > >
> > >
> > > Besides, as for your comment that suggests showing the plot of \#query vs. total training time, we remind that the offline training time cost is not influenced by the number of online queries, because in our implementation we aim to learn complete geodesics of the whole mesh model, regardless of how many queries the users may specify. This setting is reasonable according to the preceding paragraph explaining our suitable application scenarios. More importantly, even if in some cases training time cost is really the most critical consideration factor, one can have multiple choices to deal with the speed-accuracy trade-off:
> > >
> > >
> > > -- ***(a) Preparing sparser ground-truth pairs.*** As reported in Table S1 of the supplementary material as well as Table R3 of the uploaded PDF file, one can choose to generate sparser ground-truth pairs to significantly reduce the time cost of pre-processing procedures, and the resulting geodesic representation accuracy can still maintain satisfactory.
> > >
> > >
> > > -- ***(b) Early stopping of the training process.*** As can be found in our rebuttal response to your [Q1], the fast convergence speed of our NeuroGFs allows us to flexibly control the training time cost while maintaining highly competitive performances.
> > >
> > >
> > > -- ***(c) Adopting generalizable NeuroGF learning frameworks.*** As designed and comprehensively evaluated in our rebuttal (Figure R1 and Table R4 of the uploaded PDF file), one can directly use the generalizable version of our NeuroGF representation framework. After training, the generalizable NeuroGFs can be directly applied to process unseen shapes and categories while achieving satisfactory performances.

---

### Official Review · Reviewer_yBGS · 2023-07-05

**Soundness:** 3 good
**Presentation:** 3 good
**Contribution:** 1 poor
**Rating:** 3
**Confidence:** 5

**Summary:**

This paper presents a framework to effeciently compute pairwise geodestic distances and shortest geodesic paths on the surface of a given mesh. This is accomplished by overfititng a neural learning model to a given mesh to regress the distances, paths and a SDF to reconstruct the surface. The geodesic distances/paths are learned in an implicit manner given pairs of points on the surface - specifically, points are sampled along the line between the points and the corresponding points on the geodesic path are predicted by a neural network.

**Strengths:**

The problem being tackled, namely of effecient geodesic computations is important for the graphics community. The method is simple and easy to understand and reproduce.

**Weaknesses:**

1. In my opinion, the overfitting setting weakens the contribution significantly, especially considering that GraphWalks [Potamias et al] performs this task with a dataset and can predict even if with lower accuracy, on unseen test shapes. This work can be extended to learn on datasets, for eg simply by learning a global shape descriptor and appending it to the existing learned point features.

2. Since its in the overfitting sitting, the results are less impressive. Neural networks are known for the ability to interpolate between seen data - i.e, if the model is trained to predict geodesics between enough number of pairs, its not too difficult to then get a good performance on the remaining pairs at inference.

3. Additionally, since it is in the overfitting setting and thus needs to be trained for every new mesh, the training time itself should be considered as part of the time taken to compute the geodesics. The training time for each mesh is not mentioned in the paper.

4. The network takes in point coordinates (via FC layers) and thus the neighborhood of a point is not encoded in any way - a better representation would be to encode as well the immediate neighborhood of the given pair of points. However, this has not been tried in the paper.

5. The role of the SDF is simply as a byproduct, as noted by the authors in the supplemental. Table 3 shows that adding the SDF only provides a small improvement.

**Questions:**

1. The computation of the Distrubution constraint (eq 11) is not clear - the text says that a separate network is trained to predict the SDF of the generated curve poitns P_m, and that it is trained in advance and kept fixed. However, P_m itself is computed during training (from the line sampels), so how is the network trained in advance?

**Limitations:**

As discussed in the weakness section, the primary limitation is that the method has to be trained for every given mesh, thus I'd consider the training time as part of the geodesic computation time. I'd suggest the authors to extend the method to learn on datasets with a global shape descriptor to make it more useful for the community. In my opinion, the overfitting setting weakens the result and is not as useful. The work would be very useful if it can be used to perform online computations on unseen shapes.

---

> ### Author Rebuttal · Authors · 2023-08-10
>
> ### **[Rebuttal to Reviewer yBGS]**
>
>
> ### **[W1]** *Weakened contribution due to the overfitting setting; Extension to learn on datasets.*
>
> **Response:** Thanks for your insightful advice, yet we beg to differ with the judgment that the overfitting setting "significantly" weakens our contribution. In terms of the overfitting paradigm, NeuroGF inherits from previous works like SEP [13] (C. Gotsman, et al., SIGGRAPH Asia-22'), which can also be viewed as the process of "compressing geodesics". As summarized in [13], this setting well suits many interactive applications where rapid computation of arbitrary point-to-point geodesics on very large meshes is required.
>
> Still, we do agree with the reviewer's advice that it is very promising to extend NeuroGFs to be generalizable models. We made efforts to such extensions and performed comprehensive experimental evaluations. As shown in Figure R1 of the uploaded PDF file, we designed three versions of generalizable NeuroGFs using (a) *autodecoder-based*, (b) *point transformer-based*, and (c) *graph convolution-based* feature extraction strategies. Notably, NeuroGF equipped with (a) or (b) for shape encoding can directly work on point clouds during testing. These modifications are technically straightforward since we only need to replace the query point embedding stage with a feature-conditioned process.
>
> We used the popular ShapeNet mesh dataset pre-processed by DISN (Q. Xu, et al., NeurIPS-19'), which covers 13 different shape categories. We collected 3000 models from 8 categories as our training set. For each model, we only sparsely generated 2K ground-truth training pairs. For evaluation, we constructed different testing sets: 1) SN-Airplane, SN-Chair, and SN-Car are collected within the same categories of *airplane*, *chair*, and *car*, each of which containing 500 models; 2) SN-8x50 is collected from the same 8 categories as in the training set (each category with 50 models), but each shape is unseen during training. 3) SN-5x50 is collected from the other 5 different categories (each category with 50 models).
>
> Results are provided in Table R4 of the uploaded PDF file. The testing results on (a) validate the category-specific representation capability (with about 3\% MRE). The testing results on (b) show that our extended approach equipped with a powerful deep point encoder works well on point clouds for both seen and unseen categories. And (c) further incorporates mesh connectivity cues, thus achieving better performance.
>
>
> ### **[W2]** *Not difficult to obtain good performance with enough number of training pairs.*
>
> **Response:** Please refer to our detailed responses to Reviewer oZNk [W1] and [W2]. Briefly speaking, the actual ratio of "our used training pairs" to "all pairs between vertices of the original mesh" for geodesics ground-truth preparation is typically small, and our approach can still maintain relatively satisfactory performances even with much sparser (e.g., thousands of) ground-truth training pairs.
>
> ### **[W3]** *Time cost of training NeuroGF for overfitting each new mesh.*
>
> **Response:** Practically, since NeuroGF is an offline overfitting process, users can flexibly control the trade-off between training time and fitting accuracy. Typically, the training time cost for achieving comparable performances as reported in the paper is "minute-level". Below we provide statistics averaged on all the 10 testing models, showing that the convergence speed is fast. For example, it only takes less than half a minute to reach the MRE lower than 3\%.
>
> | MRE | <3\% | <2\% | <1\% |
> | :----: | :----: | :----: | :----: |
> | **Training Time** | 0.4min | 1.7min | 7.9min |
>
> From our experience, in most cases, the whole training effects will saturate within 10 minutes. Further extending the training process can only bring marginal performance gains at a slow pace.
>
> ### **[W4]** *Encoding neighborhood of given pair of query points.*
>
> **Response:** Query point encoding is indeed an important issue, as presented in our response to Reviewer Racd [Q1] and our newly-added ablation study (variant (1) in Table R1 of the uploaded PDF file) of adding NeRF-style position encoding to point coordinates. However, in our processing pipeline, the technical soundness of encoding neighborhood of query points migh be questionable. This is because, in real application scenarios, users may randomly specify query points with varying density and distribution. In an extreme case, if a user only specify a single pair of query points, the neighbor aggregation process would fail. Hence, we still adopt our original design.
>
>
> ### **[W5]** *Role of the SDF learning branch.*
>
> **Response:** Essentially, we include the SDF branch with the goal of forming a unified representation for encoding both 3D geometry and geodesics information, and it also brings actual benefits. In addition to the boosting effect (though not very significant) as verified in our ablation studies, the learning of SDF also speeds up the overall convergence speed of geodesics branches. The average training time needed for reaching MRE lower than 3\% and 2\% will respectively increase to 0.9min and 2.4min if we remove the SDF branch.
>
> ### **[Q1]** *How the separate SDF network is trained for the distribution constraint.*
>
> **Response:** It seems that there exists a misunderstanding of this learning procedure. Specifically:
> - We remind that the notation $\mathcal{N}_\phi$ used in Eq. (11) is a separate network for SDF fitting.
> - This separate SDF network is different from the notation $\mathcal{N}_\Theta$ as given in Eq. (1). It will cause confusion if mixing them up.
>
> The separate SDF network $\mathcal{N}_\phi$ is not optimized on the generated curve points $\mathbf{p}_m$. Instead, it is optimized on randomly sampled spatial queries in advance. After finishing training, we freeze its parameters and apply it to infer SDF values of the generated curve points $\mathbf{p}_m$.

---

> > ### Author Response · Authors · 2023-08-21
> >
> > Dear **Reviewer yBGS**
> >
> > Thank you for taking the time to review our submission. As the discussion phase between the reviewers and authors is coming to the end, may we know whether there are still unsolved concerns from you? We are pleased to address them. Looking forward to receiving your feedback.
> >
> > Best regards,
> >
> > The authors

---

### Official Review · Reviewer_oZNk · 2023-07-06

**Soundness:** 2 fair
**Presentation:** 3 good
**Contribution:** 2 fair
**Rating:** 4
**Confidence:** 4

**Summary:**

This paper develops a neural network architecture to estimate the geodesic distances and shortest geodesics between query points on a given 2D surface. It also provides a signed-distance function field evaluated at the given query points. The architecture consists of a set of FC layers followed by Pointwise MLPs. The learning process is supervised. That is, the output of the network is compared with the ground truth distances and geodesic paths. The paper demonstrates these ideas for a number of 3D surfaces and compares the results with two previously published approaches. The results are found to be more accurate and take less computing time.

**Strengths:**


There are many techniques in shape analysis of 3D objects that require computing geodesics or geodesic distances between arbitrary points. This paper develops a learning based solution that provides a slightly more accurate estimates and a somewhat faster pace.

**Weaknesses:**


In my opinion this problem is not that challenging. The paper uses about 20K points on a surface and generates ground truth quantities (geodesic distances, geodesics, etc) from them. This seems to me like a very dense sampling. If on a 2D surface we have pairwise distances between 20K (or some such number) points, then the task of finding distances between remaining arbitrary pairs does not seem that challenging. Indeed, the errors across different methods are not that different – mostly within a one percentage point or so.

In terms of regression, one is learning a function f: S x S --> R_+ using millions of data points for the geodesic distance estimation. How about training the algorithm it only hundreds or even thousands of paired distances.

**Questions:**

The conclusion states that the paper learn “neural implicit representations for 3D surface geodesics”. I do not quite understand what this means. Is there a mathematically precise statement that can replace this ambiguous claim.

**Limitations:**

As the authors state, there is no guarantee that the shortest path lies on the actual surface. This seems like a big limitation. How much error does the post-processing introduces in this?

---

> ### Author Rebuttal · Authors · 2023-08-09
>
> ### **[Rebuttal to Reviewer oZNk]**
>
>
> ### **[W1]** *Very dense sampling of ground-truth training data.*
>
> **Response:** There seems to be a misunderstanding about preparing ground-truth pairs for training. In our experiments, we will create a simplified version of mesh model with around 20K vertices if the original mesh is of larger scale (note that the testing data are still obtained from the original input mesh). Then as presented in the supplementary material (page 3, lines 17-19) we compute and preserve paired ground-truth geodesics between 1024 source points and 4096/2048 target points (4096 for geodesic distances, 2048 for shortest paths), rather than all pairs between the 20K vertices. The proportions of "our used training pairs" to "all pairs between vertices of the original mesh" for geodesic distance ground-truth preparation are given below. We can see that the ratio of our used pairs is very small, except for the sparse *nail* model.
>
>
> | Mesh &#124; | armadillo | bimba | bucket | bunny | cow | dragon | fandisk | heptoroid | maxplanck | nail |
> | :----: | :----: | :----: | :----: | :----: | :----: | :----: | :----: | :----: | :----: | :----: |
> | **Ratio** &#124; | 0.03% | 0.15% | 0.69% | 0.69% | 0.40% | 0.004% | 2.10% | 0.01% | 0.35% | 71.2% |
>
>
> Moreover, we did experiment with different numbers of ground-truth training data, as comprehensively evaluated in Table S1 of our supplementary material, where the second row (\#Sources=1024) corresponds to the setup we used in the paper. We can observe that the resulting performance (MRE of 2.21\% on the challenging dragon model) can still be competitive even when we only use 32 source points (the last row).
>
> ### **[W2]** *Training with highly sparse ground-truth pairs.*
>
> **Response:** To better address your concern, in addition to the experiments conducted in Table S1 of our supplementary material, we further explored much sparer settings of paired ground-truth preparation. Due to limited time, here we only used five testing meshes *armadillo*, *bunny*, *cow*, *dragon*, and *nail*, and listed the average performances in Table R3 of the uploaded PDF file. When training NeuroGFs with only 8K and 2K ground-truth pairs, the errors still maintain relatively satisfactory.
>
> Besides, please also refer to our newly-added experiments (Table R4 of the uploaded PDF file) of extending NeuroGFs to be generalizable learning models (as asked by Reviewer yBGS), where we only prepared about 2K ground-truth pairs for each training mesh.
>
> ### **[Q1]** *Ambiguous meaning of neural implicit representations for 3D surface geodesics.*
>
> **Response:** The concept of "neural implicit representation" has been popular in recent years, especially for representing 3D signals, such as DeepSDF (CVPR-19'), NeRF (ECCV-20'), and their numerous follow-up works. Generally, corresponding to "implicit representation", traditional 3D data structures like meshes, voxels, and point clouds are known as "explicit representation". That is, they have a finite number of explicitly stored data elements, like vertex/triangle, voxel occupancy status, and spatial point. Differently, neural implicit models tend to represent the target signal with infinite resolutions in a "query-and-answer" fashion. For example, given a certain 3D query point position, we feed it into a neural network, which outputs the scalar value of its signed distance. Thus, by densely sampling query points and collecting their outputs, we can flexibly reconstruct the surface geometry from the signed distance field with arbitrary resolution (depending on the number of queries).
>
> In our case, the formal mathematical description of the proposed "neural implicit representations for 3D surface geodesics" has already been given by Eq. (1) in the paper. Specifically:
> - The input query is a pair of source and target points ($\mathbf{q}_s$;$\mathbf{q}_t$) located on the underlying surface.
> - The neural model outputs a scalar value of geodesic distance $d$ and a sequence of shortest path points $\mathbf{c}_{s \rightarrow t}$.
>
> ### **[L1]** *1) No guarantee that the shortest paths lie on the surface; 2) Effects of post-processing.*
>
> **Response:** As both quantitatively and qualitatively illustrated in the paper, the generated shortest paths are close enough to the underlying iso-surface, meaning that we can conveniently deduce post-processed shortest paths whose curve points are exactly located on the surface by straightforward local projection. Here, we simply implement this process by locally sampling surface points and then performing nearest-neighbor matching for the raw outputs of curve points. As shown in Table R2 of the uploaded PDF file, such a post-processing procedure consistently brings further accuracy improvement for the prediction of shortest paths on ALL the testing meshes. On average, the error decreases from the original $1.25 \times 10^{-2}$ to $1.09 \times 10^{-2}$. Hence, we beg to differ with the judgment that "This seems like a big limitation".

---

> > ### Author Response · Authors · 2023-08-21
> > **Looking forward to receiving your feedback**
> >
> > Dear **Reviewer oZNk**
> >
> > Thank you for taking the time to review our submission. As the discussion phase between the reviewers and authors is coming to the end, may we know wether there are still unsolved concerns from you? We are pleased to address them. Looking forward to receiving your feedback.
> >
> > Best regards,
> >
> > The authors

---

### Official Review · Reviewer_Racd · 2023-07-10

**Soundness:** 3 good
**Presentation:** 3 good
**Contribution:** 2 fair
**Rating:** 5
**Confidence:** 4

**Summary:**

This paper proposed to employ neural geodesic fields to implicitly represent (1) geodesic distance, (2) signed distance filed, (3) shortest geodesic path query using the overfit-paradigm. Experiments are conducted to demonstrate the effectiveness of the proposal.

**Strengths:**

• The writing of this paper is clear and easy to follow.
• The extension to use  implicit neural representation to represent geodesic distance and efficiency query is novel. It is a simple and yet seems effective approach under the overfitting paradigms for geodesic implicit representation.



**Weaknesses:**

1. The diversity of the mesh shape used in the experiments is limited. No realistic real-world mesh is experimented. The original scale of mesh shape in the experiments is not diverse enough: e.g. from small toy scale to large skyscraper scale mesh to validate the proposal in a more realistic setting.




**Questions:**

1. Page 4, section 3.2, any ablation study on input points embedding? Any comparison regarding FC layers vs position embeddings for point coordinates? Usually position embedding might work better for avoiding over-smoothing of the input signals for complicated shapes. It might worth exploring.
2.  Section 3.3, any ablation studies on the learning objectives? Why L1? How about others?
3. Table 1, please annotate the best and worst performing entries in comparison.
4. Section 4.3, more ablation analysis is needed. For example, more conclusive analysis is needed for Table 3.

---

> ### Author Rebuttal · Authors · 2023-08-09
>
> ### **[Rebuttal to Reviewer Racd]**
>
> ### **[W1]** *Not enough shape diversity and scale of the used testing meshes.*
>
>
> **Response:** We would like to remind that some of the testing models used in our experiments (such as *armadillo*, *bunny*, *dragon*) are real-world meshes, which are created from real-scans. Please have a check from the website of *The Stanford 3D Scanning Repository*.
>
> Still, we do agree with the reviewer that it is valuable to perform evaluations on more diverse mesh models. To this end, here we considered both *1) data volume scale* and *2) scene representation scale*.
>
> For the former, we experimented with a much denser version of the *dragon* model with 1.5M vertices and another classic graphics model *lucy* (which is also created from real scans) with 6.9M vertices. As reported in the Table below, our approach works well on these two highly dense mesh models.
>
>
> | Mesh | #V (M) | MRE (%) | Chamfer-$L_1$ ($\times 10^{-2}$) |
> | :----: | :----: | :----: | :----: |
> | dragon | 1.5 | 0.70 | 1.335 |
> | lucy | 6.9 | 0.58 | 1.404 |
>
> For the latter, we further performed evaluations on three large-scale real-scanned meshes covering (a) indoor room, (b) urban scene, and (c) street view, which are from diverse dataset sources. Since these realistic scene scans are highly messed up, we turn to use the classic Dijkstra algorithm to compute geodesics as training and testing data. As shown in Figure R2 of the uploaded PDF file, our approach still achieves satisfactory performances (with MRE rates of 0.37\%, 1.14\%, and 0.23\%, respectively) on the three scene-level meshes.
>
> ### **[Q1]** *Ablation study on the position encoding of query point coordinates.*
>
> **Response:** Thanks very much for your valuable suggestions. Accordingly, as reported in Table R1 of the uploaded PDF file, we conducted the corresponding ablation study by applying a classic position encoding operation (as used in NeRF, ECCV-20') to input query point coordinates before the subsequent FC layers for high-dimensional feature embedding. Indeed, we can observe that the resulting performances on the two complex meshes *dragon* and *heptoroid* are improved in different degrees, demonstrating the potential of exploring more advanced position encoding strategies.
>
> ### **[Q2]** *Ablation study on the distance metric used to formulate the learning objectives.*
>
> **Response:** Thanks very much for your detailed evaluation of our specific choices of technical implementation. In fact, we did experiment with some different ways of formulating the learning objectives in terms of the choice of distance metric. For example, we can choose to use $L_2$ loss instead of $L_1$ to calculate all learning objectives. Besides, for the formulation of shortest path supervision (i.e., $\ell_\mathrm{spath}$ formulated as Eq. (8) in the paper), we can also choose the popular Chamfer distance to supervise the curve deformation process. The resulting performances obtained from such two different implementation variants are reported in Table R1 of the uploaded PDF file. We can observe that in most cases using $L_2$ loss causes different degrees of performance degradation, and Chamfer distance leads to further performance improvement. Still, in our implementation, we did not adopt Chamfer distance as our final choice because of its additional computational burden and memory cost.
>
> ### **[Q3]** *Annotating the best and worst performing entries in Table 1 of the paper.*
>
> **Response:** Thanks very much for your useful suggestion to help improve the formatting quality of our manuscript. We will accordingly revise Table 1 in the final version.
>
> ### **[Q4]** *Enriching conclusive analyses in Section 4.3 of the paper.*
>
> **Response:** Thanks very much for your valuable comments, and we are sorry for not being able to place detailed ablative analyses in the paper due to page limits. According to your suggestion, we will supplement more adequate and in-depth analyses to facilitate readers' understanding of our approach in the supplementary material.

---

> > ### Comment · Reviewer_Racd · 2023-08-20
> >
> > Thank you for taking time to respond to my review. Please keep improving the paper. I will keep my original ratings for now.

---

> > > ### Author Response · Authors · 2023-08-21
> > >
> > > Dear Reviewer Racd
> > >
> > > Thanks very much for your time and efforts in evaluating our rebuttal contents. We will supplement these newly-added discussions and experimental results into the final version to further improve the quality of our paper.
> > >
> > > Best regards,
> > >
> > > Authors

---

### Author Rebuttal · Authors · 2023-08-09

### **[Global Response]**

We sincerely thank all five reviewers for their time and efforts in reviewing this paper and providing different aspects of valuable suggestions and helpful comments. In summary, during the rebuttal period, we made the following major efforts to address reviewers' concerns. The critical contents and results will be included in our paper or supplementary material to further improve the quality of our paper.

- We experimented with additional testing meshes with more diverse types and larger scales.

- We conducted more ablation studies to explore different choices of our specific technical implementations.

- We demonstrated the effectiveness of our NeuroGF when trained with a limited amount of ground-truth pairs.

- We introduced the post-processing procedures for making the generated shortest path points lie exactly on the underlying surface, which can further boost the representation accuracy.

- We conducted extensions of generalizable NeuroGF learning frameworks and evaluated their effectiveness on the popular and widely-used ShapeNet dataset.

- We clarified/presented the statistics of network training (as well as other pre-processing procedures) time cost.

- We explained the goal, necessity, and effects of adding the auxiliary SDF learning branch.

- We carefully answered reviewers' questions about some confusions/misunderstandings of our approach and some setups/procedures to facilitate better assessments of this work.

For convenience, below we will briefly summarize each raised Weakness (W), Question (Q), and Limitation (L), and provide our response item by item. Note that there is also a one-page PDF file uploaded for presenting more figures and tables.

---

> ### Author Response · Authors · 2023-08-16
> **Looking forward to hearing from you**
>
> Dear Reviewers **Ujtd**, **yBGS**, **oZNk**, and **Racd**,
>
> Thank you for taking the time to review our submission and providing us with constructive comments. We would like to know if our responses adequately addressed your earlier concerns. Additionally, if you have any further concerns or suggestions, we would be more than happy to address and discuss them to enhance the quality of the paper. We eagerly await your response and look forward to hearing from you.
>
> Best regards,
>
> The authors

---

### Decision · Program_Chairs · 2023-09-21

**Decision:**

Accept (poster)

**Comment:**

The paper introduces an efficient method for estimating geodesic paths on 3D meshes using neural implicit functions. All the reviewers appreciated the straightforward method and acknowledged the relevance of the problem, together with a notable efficiency and the good experimental results that are reportedly competitive with respect to prior (and extensive) art. Major weaknesses were identified in the limited diversity of shapes and on the projection step, required for the estimated paths to lie on the surface. The authors provided a rather extensive rebuttal, including several additional shapes, more ablation studies, more datasets, additional evaluation and discussion addressing all the above points. The AC also read through the paper, acknowledging the previously raised points, as well as these being satisfactorily addressed by the rebuttal. We therefore recommend acceptance, while strongly recommending the authors to incorporate all the new material in the final revision of the paper.